# Genome-Wide Association Study of Resistance to Largemouth Bass Ranavirus (LMBV) in *Micropterus salmoides*

**DOI:** 10.3390/ijms251810036

**Published:** 2024-09-18

**Authors:** Pinhong Li, Xia Luo, Shaozhi Zuo, Xiaozhe Fu, Qiang Lin, Yinjie Niu, Hongru Liang, Baofu Ma, Ningqiu Li

**Affiliations:** 1Pearl River Fisheries Research Institute, Chinese Academy of Fishery Sciences, Key Laboratory of Fishery Drug Development, Ministry of Agriculture and Rural Affairs, Guangdong Provincial Key Laboratory of Aquatic Animal Immunology and Sustainable Aquaculture, Guangzhou 510380, China; d220100042@st.shou.edu.cn (P.L.); lxwenhao@163.com (X.L.); shaozhi.zuo@pirbright.ac.uk (S.Z.); fuxiaozhe-1998@163.com (X.F.); lin9902057@163.com (Q.L.); niuyinjie0530@163.com (Y.N.); hrliang13@126.com (H.L.); mabf@prfri.ac.cn (B.M.); 2College of Fisheries and Life Science, Shanghai Ocean University, Shanghai 201306, China

**Keywords:** *Micropterus salmoides*, largemouth bass ranavirus (LMBV), genome-wide association analyses (GWAS), single nucleotide polymorphisms (SNPs), insertions–deletions (InDels)

## Abstract

The disease caused by Largemouth bass ranavirus (LMBV) is one of the most severe viral diseases in largemouth bass (*Micropterus salmoides*). It is crucial to evaluate the genetic resistance of largemouth bass to LMBV and develop markers for disease-resistance breeding. In this study, 100 individuals (45 resistant and 55 susceptible) were sequenced and evaluated for resistance to LMBV and a total of 2,579,770 variant sites (SNPs-single-nucleotide polymorphisms (SNPs) and insertions–deletions (InDels)) were identified. A total of 2348 SNPs-InDels and 1018 putative candidate genes associated with LMBV resistance were identified by genome-wide association analyses (GWAS). Furthermore, GO and KEGG analyses revealed that the 10 candidate genes (*MHC II*, *p38 MAPK*, *AMPK*, *SGK1*, *FOXO3*, *FOXO6*, *S1PR1*, *IL7R*, *RBL2*, and *GADD45*) were related to intestinal immune network for IgA production pathway and FoxO signaling pathway. The acquisition of candidate genes related to resistance will help to explore the molecular mechanism of resistance to LMBV in largemouth bass. The potential polymorphic markers identified in this study are important molecular markers for disease resistance breeding in largemouth bass.

## 1. Introduction

Largemouth bass (*Micropterus salmoides*) is one of the most economically productive freshwater fish species in the world. According to the data reported by the Food and Agriculture Organization (FAO), the global production of largemouth bass in 2022 exceeded 804,000 tons [1]. This indicates that the largemouth bass is one of the fish with great economic value. However, since the outbreak of largemouth bass ranavirus disease (LMBVD), the largemouth bass aquaculture industry has suffered severe economic losses [2,3]. The onset temperature of LMBVD ranges from 25 °C to 32 °C, and dramatic changes in environmental factors and intensive feeding density lead to increases in the mortality rate [4,5]. Some detecting methods for LMBV were developed, including conventional PCR, qPCR, LAMP, ELISA, and AuNPs [6,7,8,9,10]. Studies on the prevention and control of LMBV mainly focused on the development of vaccines and drugs [11,12,13]. Up to now, there are no safe and effective drugs to prevent or control LMBV infection. In the context of huge market demand and the impact of diseases on the development of the largemouth bass industry, breeding disease-resistant strains has become the main goal of the largemouth bass industry.

Selective breeding, especially genome-based selective breeding, is considered an effective strategy to improve the disease resistance in fish [14,15]. The development of next-generation sequencing (NGS) technology and computational tools helps to discover single nucleotide polymorphisms (SNPs) and insertions–deletions (InDels) and genotyping in whole genomes [16]. SNPs and InDels are genomic differences and variations that can have a significant impact on the biological and physical characteristics of organisms. SNPs-based markers are often used to identify quantitative trait loci (QTL) or candidate genes related to target traits [17]. Genome-wide association studies (GWAS) have been widely used to identify SNPs associated with important economic traits such as growth and disease resistance [18,19,20]. However, there is still limited research on using InDels for GWAS in fish.

Disease resistance traits are particularly important for aquatic animal breeding because they directly determine the survival rate and yield of aquaculture species. GWAS has been reported to identify SNPs associated with disease resistance traits in species of significant commercial value in aquaculture, including Atlantic salmon (*Salmo salar* L.) [20], rainbow trout (*Oncorhynchus mykiss*) [21,22], sea bass [23,24], Large yellow croaker (*Larimichthys crocea*) [25], catfish [26], *Cynoglossus semilaevis* [27], Tambaqui (*Colossoma macropomum*) [28], Nile tilapia (*Oreochromis niloticus*) [29], and *Paralichthys olivaceus* [30]. The discovery of genetic markers associated with disease resistance has facilitated the application of molecular marker-assisted breeding (MAB) to increase the disease resistance of various aquaculture species. For example, researchers have successfully developed a Japanese flounder population that is resistant to lymphocystis disease, as well as a novel strain of red sea bream (*Pagrus major*) that exhibits resistance to Red Sea bream iridovirus disease (RSIVD), by employing marker-assisted selection techniques [31,32]. However, to date, molecular markers associated with disease resistance traits in largemouth bass have not been identified. Therefore, it is important to study the genetic markers associated with LMBV resistance in largemouth bass.

The candidate genes related to disease resistance traits help to elucidate the interaction mechanism between hosts and pathogens. Several candidate genes related to the immune response have been discovered through GWAS, such as Sentrin-specific protein 5 (SENP5) in rainbow trout [21], lysine-specific metabolism 2A (KDM2A), beta defensin 1 (DEFB1), and cystatin B (CSTB) in Asian Seabass [24], as well as galectin-7 (LGALS17), Vacuolar protein sorting-associated protein 52 (VPS52), and Tripartite motif-containing protein 29 (TRIM29) in tilapia [29].

The release of genome sequence information on largemouth bass laid the foundation for the exploration of variant sites and candidate genes for disease resistance [33]. In this study, we used whole genome re-sequencing to genotype largemouth bass samples challenged by LMBV infection. And the SNPs-InDels and candidate genes significantly associated with resistance to LMBV were identified by GWAS. The functions of these candidate genes and the signaling pathways were further analyzed. The genetic markers obtained by GWAS in this study contribute to the molecular mechanisms of host antiviral resistance and the development of molecular marker-assisted selection in largemouth bass.

## 2. Results

### 2.1. Experimental Challenge and Samples

A total of 1325 fish were used for the challenge experiment. Throughout the entire period of LMBV infection, clinical signs related to the viral infection were observed, including blackening of body color and abnormal swimming. After the challenge experiment, a total of 1074 dead individuals (susceptible populations) and 251 surviving individuals (resistant populations) were obtained, with an overall mortality rate of 81.06%. In total, 100 samples, composed of 45 resistant and 55 susceptible individuals, were selected for subsequent resequencing analysis. The binary survival status (0 for fish that survived during the challenge period and 1 for fish that died during the 14-day post-challenge observation period) of the individuals was recorded.

### 2.2. SNP Data and Density on Chromosomes

A total of 100 genome resequencing libraries were sequenced. We uploaded the raw sequence data to the NCBI database (https://www.ncbi.nlm.nih.gov/, (accessed on 19 August 2024)), with the accession number of PRJNA1144216. After preliminary quality control, an average of 10,556,429,233 bp high-quality data were obtained for each sample. The ratio of high-quality bases (Q score > 30) was 92.48% and the average GC content was 42.45%. By comparing with the reference genome, a total of 3,571,046 SNPs-InDels were obtained. Further quality control filtering was performed, resulting in 2,579,770 high-quality SNPs-InDels. The distribution of high-quality SNPs on 22 chromosomes was shown in Figure 1; the SNP density map showed that density was highest on chr 1 and lowest on chr 15.

### 2.3. Analysis of the Linkage Disequilibrium Population Structure

The genotype data of the samples were plotted against physical distance (kb) for LD decay to reveal non-random associations of alleles at two or more loci in the population. The LD analysis (Figure 2A) showed that there was a rapid decrease in linkage disequilibrium (LD) between genetic markers. It was observed that the LD coefficient (r^2^) decreased by 50% at a distance of 25 kb. The kinship showed that the genetic relationship co-efficient was between −0.1 and 0.1 (Figure 2B,C), which indicated weak genetic relationships among the experimental population. The phylogenetic tree analysis showed that the population in this experiment is divided into three branches (Figure 2D).

### 2.4. GWAS for LMBV Resistance

The significance threshold was determined using Bonferroni correction, which sets the threshold as *p*-value = (0.05/N) where N is the number of SNPs-InDels. In this study, the significance threshold was −log10 (0.05/2,579,770) = 7.71. However, there were no SNPs identified using this threshold that were significantly associated with LMBV resistance. Therefore, we adjusted the significance threshold to −log10 *p* = 4 (Figure 3A) [34,35]. We performed GWAS using high-quality SNPs-InDels and phenotypic data through the GLM model, and finally used Manhattan plots to visualize loci with significant associations (−log10 *p* = 4) (Figure 3A). A total of 2348 variants (1925 SNPs and 423 InDels) significantly associated with LMBV resistance were obtained by GWAS. Among the number of SNP mutation types, there are 32 SNPs with non-synonymous mutations and 58 SNPs with synonymous mutations (Appendix A). Among the number of InDels, four mutations occurred in exonic (Appendix A). It is worth mentioning that all significant markers associated with LMBV resistance traits were widely distributed on almost all chromosomes. Among them, most of the significant variant sites were located on chromosome 3 with a number of 1726. The QQ plot of the *p*-values is shown in Figure 3B. We observed that the *p*-values of the significant SNPs-InDels (−log10 *p* = 4) were larger than their expected values, indicating that the GLM model was reliable for the sample data. According to the *p*-values, the top 10 SNPs-InDels associated with resistance to LMBV in *Micropterus salmoides* are shown in Table 1.

### 2.5. GO/KEGG Enrichment of Candidate Genes

After annotating the upstream and downstream 25 kb genome sequences based on significant variant sites, we found a total of 1018 candidate genes associated with LMBV resistance. These genes were subjected to GO/KEGG enrichment analysis. According to the GO enrichment analysis, candidate genes are involved in biological processes mainly related to cellular processes (Figure 4A). Candidate genes are primarily enriched in cellular components, such as the cell and cell parts. Additionally, they are predominantly enriched in molecular functions, particularly those related to binding and catalytic activities. (Figure 4A). KEGG analyses revealed that one candidate genes were related to the intestinal immune network for the IgA production pathway (MHC II) (Figure 4B). Nine candidate genes were enriched in the FoxO signaling pathway (*p38 MAPK*, *AMPK*, *SGK1*, *FOXO3*, *FOXO6*, *S1PR1*, *IL7R*, *RBL2*, and *GADD45*) (Figure 4B).

## 3. Discussion

LMBV is one of the most serious threats to the largemouth bass aquaculture industry. Currently, there are no effective drugs for preventing and treating LMBV infection. Genetic and genomic studies of disease resistance in aquatic animals are becoming more widespread, and many study cases of genetic variation in disease resistance by GWAS have been published [21,24,30]. Improving the overall resistance of the population to LMBV is beneficial for the sustainable development of the largemouth bass aquaculture industry. In this study, we performed a GWAS to identify SNPs-InDels and candidate genes related to LMBV resistance for the selection of LMBV-resistant strains.

Measuring disease resistance phenotypes is an important hurdle in selecting for disease resistance [36]. In fish, disease resistance phenotypes are often obtained through survival data challenged by field tests or pathogen challenge experiments [23,29]. Therefore, in order to obtain the disease-resistant phenotype, we conducted LMBV challenge experiments on largemouth bass. In addition, we referred to the records of flounder VHSV-resistance traits [30] and obtained the binary survival phenotype data of LMBV-resistance. Apart from the phenotype, genotype identification is another necessary factor for conducting GWAS [17]. We applied next-generation sequencing (NGS) technology to genotype samples of largemouth bass. In order to eliminate the influence of population structure on false positives in GWAS analysis results, LD and Kinship analyses were conducted in this study. The results showed that the population used in this study exhibited LD decay and had weak kinship; the gene tree of the population showed that the population could be divided into three branches, suggesting the reliability of the GWAS results.

Numerous studies have demonstrated that disease resistance traits in aquaculture species are the result of polygenic effects. In *O. niloticus*, 29 SNPs associated with Tilapia lake virus (TiLV) resistance traits were detected on chromosomes 3 and 22 [29]. In *Dicentrarchus labrax*, three QTLs related to Viral nervous necrosis (VNN) resistance traits were identified on chromosomes 3, 20, and 25 [23]. Similarly, 10 QTLs associated with infectious hematopoietic necrosis virus (IHNV) resistance traits were detected in rainbow trout [22]. However, there is still limited research on using InDels for GWAS in fish. SNPs might play an important role in resisting pathogens by affecting antiviral gene expression or protein function. The G protein-coupled receptor 143 (GPR143) was found to be present as a SNP associated with RGNNV resistance, and the expression of GPR143 showed significant differences between resistant and susceptible fish [37]. SNPs that occur in genes related to immunity, such as those encoding cytokines, may alter the host’s immune response to a virus. In fish, interleukin-6 (IL-6) is a very important immune-regulatory cytokine that plays a crucial role in host defense against viral infections. The SNP locus in the IL-6 coding region was found to be significantly associated with ISKNV resistance [38]. In this study, we detected 2348 SNPs-InDels associated with LMBV resistance. The presence of these loci may affect the function of host antiviral genes or affect the host’s immune response to Ranavirus. These assumptions need to be further verified. Furthermore, the significant SNPs-InDels were identified to be associated with LMBV resistance, which proved that resistance against LMBV in largemouth bass is controlled by a large number of loci with small effects. These loci related to LMBV resistance can provide a theoretical basis for marker-assisted breeding for disease resistance in largemouth bass.

In this study, a total of 1018 candidate genes were annotated based on significantly associated SNPs-InDels. Further analysis using KEGG enrichment revealed 10 candidate genes related to the immune response, including *MHC II*, *p38 MAPK*, *AMPK*, *SGK1*, *FOXO3*, *FOXO6*, *S1PR1*, *IL7R*, *RBL2*, and *GADD45*. Classic MHC molecules, including MHC class I and II molecules, have important immune functions in antigen presentation [39]. MHC molecules have been reported to be associated with disease resistance traits in several fish species, such as RSIV resistance in sea bream [40], PMCV resistance in Atlantic salmon [41], and *Vibrio anguillarum* resistance in Japanese flounder [42]. The association between MHC II and LMBV resistance in largemouth bass discovered in this study further indicated that MHC molecules have an important role in fish disease resistance. Our results showed that nine candidate genes were enriched in the FoxO signaling pathway. The FoxO signaling pathway regulates multiple biological processes, including cell proliferation, apoptosis, and antioxidant stress in mammals [43,44,45]. The involvement of candidate genes enriched in the FoxO pathway in immune-related signal transduction or their role in viral infection has been reported. The infection of SFTSV, HSV-1, and SARS-CoV-2 induced p38 activation and the p38 inhibitor SB203580 inhibited viral replication [46]. As a protein kinase, AMPK inhibited IFN expression, thereby promoting the replication of CyHV-3 [47]. Similarly, SGK1 stimulated viral replication following stressful stimuli [48]. FOXO3/6, as a structurally and functionally conserved transcription factor (TF), participates in signaling pathways by regulating the expression of downstream genes [45]. In *Litopenaeus vannamei*, FOXO plays an important role in antiviral immune defense by positively regulating AMPs [49]. It has been reported that overexpression of S1PR1 resulted in increased expression of the inflammatory factor but had no significant effect on viral replication [50]. As an IL-7 receptor, IL-7R is essential in B lymphopoiesis and is involved in JAK/STAT and Src activation, the PI3K/Akt pathway, and the MAPK/ERK pathway [51]. In triploid crucian carp, RBL2 was a potential marker of aging in tail fin cells [52]. RBL2 and c-myc were essential for HOXB9 to inhibit pancreatic cancer cell proliferation [53]. However, there is limited information on the role of RBL2 in viral infections. GADD45 has been shown to inhibit JNK signal transduction by targeting MKK7 [54]. In addition, GADD45 has been reported to act as a key signal transduction factor linking the NF-κB and MAPK cascades [55]. GADD45 has also been shown to reduce HIV-1 production by repressing transcription from the HIV-1 LTR promoter [56]. Although functional studies of these candidate genes have not been reported in largemouth bass, they have been reported in other species to be primarily associated with host immune responses and anti-viral infections. Therefore, the molecular mechanisms of these 10 candidate genes for LMBV resistance in largemouth bass remain to be further investigated.

## 4. Materials and Methods

### 4.1. Fish and LMBV Challenge

The animal experiment design has been approved by the Ethics Committee for Laboratory Animals of the Pearl River Fisheries Research Institute, Chinese Academy of Fishery Sciences, with the ethics approval number LAEC-PRFRI-2022-03-50. The largemouth bass “Youlu No. 3” population, which is characterized by fast growth and easy domestication, was purchased from Foshan City, Guangdong Province, China. Largemouth bass ranavirus (LMBV) was isolated and stored in our laboratory. A total of 1325 healthy largemouth bass with an average weight of about 14.50 g and a body length of about 8.90 cm were selected for the viral infection experiment. The experimental challenge was performed by intraperitoneal injection with a volume of 0.1 mL/fish in a concentration of 10^5.25^ TCID_50_/mL. In the control group, 140 fish of the same size as the experimental group were injected intraperitoneally with 0.1 mL/fish of PBS. The control group of fish and the experimental group were kept consistent under experimental conditions (such as feeding environment, water temperature, etc.) to ensure the reliability of the experimental results. After the experimental challenge, the fish were grouped and reared in a 0.25 m^3^ freshwater recirculation system at a temperature of 28 °C. During the experiment, dead fish were collected every 2 h and their fin tissue was collected and stored at −80 °C for genomic DNA extraction. At the end of the challenge, the fin samples from surviving fish were also collected and stored. Clinical signs of the disease were observed daily for 14 days after LMBV infection. Viral genomic copies and the binary survival status of each fish were recorded. The viral genomic copy number was determined by qPCR, as described previously [13]. There were two categories of resistance phenotypes: 0 for fish that survived during the challenge period and 1 for fish that died during the 14-day post-challenge observation period. Records of resistance phenotypes were used in the GWAS model analyses described below.

### 4.2. Variant Identification and Annotation

A total of 100 samples of genomic DNA were extracted and at least 3 µg of genomic DNA was used to construct paired-end libraries with an insert size of 300–400 bp using a PairedEnd DNA Sample Prep kit (Illumina Inc., San Diego, CA, USA). Sequencing was carried out on the Novaseq 6000 platform. The raw data are filtered with FASTP software (v0.19.4) to remove low-quality bases [57]. Filtered clean reads are used for assembly analysis. To identify SNPs, the BWA tool [58] was used to align the reads from each sample against the *Micropterus salmoides* reference genome (GenBank accession number: GCA_014851395.1). Variant calling was performed for all samples using the GATK software (v4.0) [59]. To determine the physical location of each SNPs-InDels, variant sites were compared and annotated using the software tool ANNOVAR (v20191024) [60].

### 4.3. Linkage Disequilibrium and Population Structure Analysis

Linkage disequilibrium (LD) is fundamental to association analysis in GWAS, determining the linkage between loci by analyzing the deviation of haplotype frequencies from expected frequencies. Through LD analysis, SNPs-InDels and candidate genes associated with resistance to ranavirus can be identified. To evaluate the LD pattern, we estimated the squared allele frequency correlation (r^2^) using PopldDecay software (v3.41) [61]. Based on the selected markers, kinship analysis was performed using GCTA software (v1.92.2) to obtain a kinship matrix between the samples [62]. The phylogenetic tree is constructed to reveal the clustering relationships between individuals by a neighbor-joining method using the software MEGA-X (v7) [63]. The bootstrapped confidence interval is based on 100.

### 4.4. Genome-Wide Association Study (GWAS)

Prior to GWAS analyses, the genotype dataset was further quality-controlled and filtered using PLINK software (v1.9) [64]. With a second allele frequency (MAF) of less than 0.05, a deletion rate of more than 0.5, and severe deviations from the Hardy–Weinberg equilibrium were excluded from further analysis. Furthermore, association analysis was conducted with the generalized linear models (GLM) using the GEMMA software (v 0.94) [65]. Manhattan plots and quantile-quantile (QQ) plots were drawn using R (v2.2.7) to visualize the results of the association analysis. The −log10 (*p*), which indicates the significance threshold, was set to 4. For each SNP, the probability *p*-value is used to measure the degree of association of the marker genotype with the phenotype; the smaller the *p*-value, the more likely the marker is to be associated with the trait.

### 4.5. Annotation of Candidate Genes

Gene enrichment analysis provided valuable clues of pathways involved for their potential biological functions. In order to identify putative candidate genes associated with LMBV resistance, we mapped the significant SNPs-InDels to the genomic sequence of largemouth bass (GenBank accession number: GCA_014851395.1). We scanned genomic regions located within a 50 kb window size (25 kb upstream and 25 kb downstream) of significant SNPs for candidate genes. To further understand the functions of candidate genes, we used the AmiGO 2 (https://amigo.geneontology.org/amigo/landing, (accessed on 19 October 2023)) to perform gene ontology (GO) analysis. We used the KEGG PATHWAY Database (https://www.genome.jp/kegg/, (accessed on 19 October 2023)) to conduct the Kyoto Encyclopedia of Genes and Genomes (KEGG) analysis. The 20 enrichment pathway analysis results of KEGG were visualized.

## 5. Conclusions

In this study, 2348 significant SNPs-InDels and 1018 potential candidate genes were found to be related to LMBV resistance of largemouth bass (*Micropterus salmoides*). Specifically, 10 candidate genes were involved in the intestinal immune network for IgA production pathway and FoxO signaling pathway. The results of this study provide a basis for further studies of the genetic structure of LMBV resistance in largemouth bass and provide valuable information for the development of molecular markers that can contribute to disease-resistance breeding programs.

## Figures and Tables

**Figure 1 ijms-25-10036-f001:**
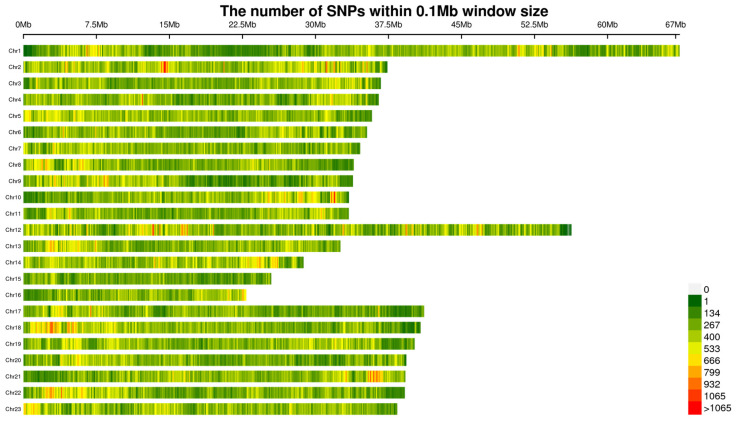
Density and distribution of the passed-filtered SNPs genotyped on each chromosome of *Micropterus salmoides*.

**Figure 2 ijms-25-10036-f002:**
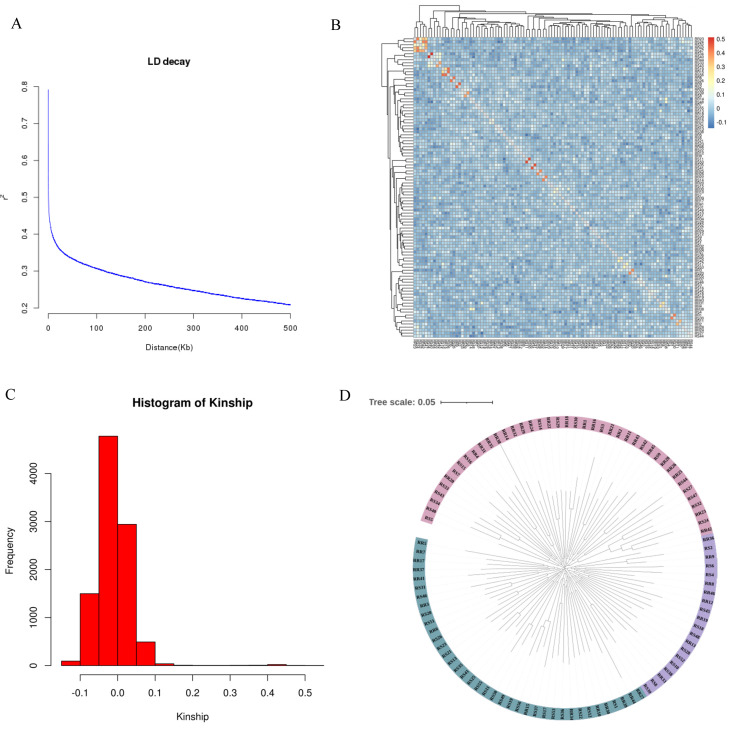
Analysis results of the population structure of experimental subjects. (**A**) The decay of LD. The horizontal coordinate is the physical distance (kb) and the vertical coordinate is the r^2^ value. (**B**) Heat map of phylogenetic relationships between samples. Red means high kinship and blue means low kinship. (**C**) Frequency distribution map of phylogenetic relationships between samples. The frequency distribution map of kinship shows the main distribution range of the values of kinship relationships between samples. (**D**) Genotype clustering analysis of experimental subjects. The tree was constructed by the neighbor-joining method, using MEGA X (v7). Evaluate the stability of each node through bootstrap testing with 100 replicates. The scale bar is 0.05.

**Figure 3 ijms-25-10036-f003:**
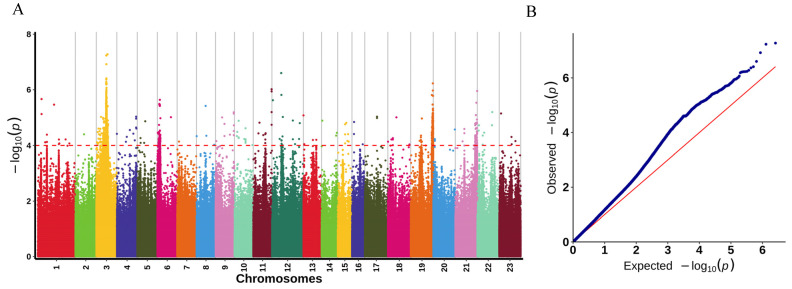
Manhattan plot of association analysis to LMBV resistance in *Micropterus salmoides*. (**A**) Manhattan plot constructed for all variant sites. The dotted red line indicates the significance threshold (−log10 *p* = 4.0). (**B**) QQ plot of the GWAS. The red diagonal line represents the expected distribution of −log10 (*p*), where *p*-values are the probabilities of observing the genetic associations by chance. The blue dots represent the observed −log10 (*p*) from the GWAS results, which correspond to the actual statistical significance of the genetic variants tested.

**Figure 4 ijms-25-10036-f004:**
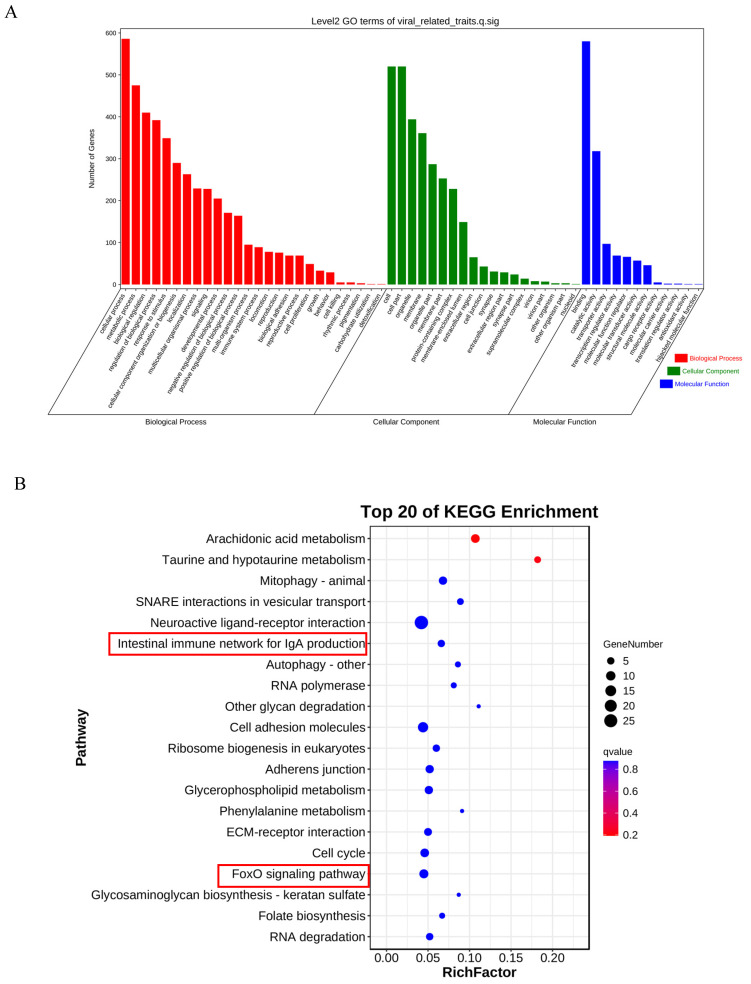
GO/KEGG enrichment analysis of candidate genes. (**A**) The bar graph of GO items, including biological process, cellular compound, and molecular function. (**B**) The bubble diagram of the top 20 KEGG pathway. Bubble area size represents the number of genes in the target gene set that belong to this pathway; bubble color represents the *q* value for the enriches significance. The red box displays the immune-related pathways enriched with candidate genes.

**Table 1 ijms-25-10036-t001:** Top 10 SNPs-InDels associated with resistance to LMBV in *Micropterus salmoides*.

Variation	Chromosome	Positon	−log10 (*p*)	Beta	PVE *
SNP	Chr3	20,046,158	7.273655921	−0.526368	0.478266619
Deletion	Chr3	17,892,358	7.231973449	−0.6084425	0.266545639
SNP	Chr3	17,962,533	6.922432256	−0.5461026	0.251943056
SNP	Chr12	16,554,664	6.602898119	0.3744209	0.277689232
SNP	Chr3	18,064,341	6.409210533	−0.5518117	0.228920207
SNP	Chr3	17,856,093	6.366583057	−0.4706976	0.21973947
SNP	Chr3	17,752,374	6.27981009	−0.503464	0.229345084
SNP	Chr3	18,074,846	6.238069156	−0.4931315	0.234229701
SNP	Chr19	39,488,902	6.230333979	−0.3551792	0.224828565
SNP	Chr3	18,327,881	6.227981378	−0.5180217	0.234803171

* PVE, proportion of variation explained.

## Data Availability

The raw sequencing data were deposited in the National Center for Biotechnology Information database (https://www.ncbi.nlm.nih.gov/) with BioProject number “PRJNA1144216”.

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
