# Peer review of "Genome-Wide Association Study of Resistance to Largemouth Bass Ranavirus (LMBV) in Micropterus salmoides"

_ijms, 2024, doi:10.3390/ijms251810036_

Round 1

Reviewer 1 Report

Comments and Suggestions for Authors

In this study, Li et al. evaluate the genetic resistance of Micropterus salmoides to Ranavirus by web experiment. They have obtained 1325 fish and corresponding virus, and sequenced 100 genomes. They then identified resistant SNP between 45 resistant and 55 susceptible individuals. Their work may be acceptable to publish in our journal, however, some revisions were needed to make it better.

Comments in this manuscript:

1. please provide the raw sequenced data of Micropterus salmoides in public website, otherwise the novelty of this manuscript will not be sufficient.

2. please provide the detailed list of some important SNPs in Micropterus salmoides

3. How many these SNP mutations are non-synonymous mutation?

4. Does the Linkage Disequilibrium have an influence on the resistance to Ranavirus?

5. Please disscuss the mechanism of SNP mutation and the resistance to Ranavirus

Author Response

Comments 1: please provide the raw sequenced data of Micropterus salmoides in public website, otherwise the novelty of this manuscript will not be sufficient.

Response 1: Thank you for your thorough review of our manuscript and for the valuable feedback provided. We acknowledge the importance of providing the raw sequence data to ensure the transparency and reproducibility of our research. Following your suggestion, we have now uploaded the raw sequence data for Micropterus salmoides to the NCBI database, with the accession number PRJNA1144216. We have included this information with a yellow background in the Results section of the revised manuscript on page 2, “2.2. SNP Data and Density on Chromosomes”.

Comments 2: please provide the detailed list of some important SNPs in Micropterus salmoides.

Response 2: Thanks for your kindly suggestions. We have compiled a detailed list of some important SNPs identified in our study. The list includes the variation type, genomic location, p-value, beta and PVE (proportion of variation explained). We have included this information with a yellow background in the Results section of the revised manuscript on page 5, “2.4. GWAS for LMBV Resistance”.

Comments 3: How many these SNP mutations are non-synonymous mutation?

Response 3: Thanks for your professional suggestions. Upon re-evaluation of our data, we found 32 SNPs non-synonymous mutations and 58 SNPs with synonymous mutations. We have included this information with a yellow background in the Results section of the revised manuscript on page 4, “2.4. GWAS for LMBV Resistance”.

Comments 4: Does the Linkage Disequilibrium have an influence on the resistance to Ranavirus?

Response 4: Thanks for your kindly suggestions. Linkage disequilibrium (LD) is fundamental to association analysis in Genome-Wide Association Studies (GWAS), determining the linkage between loci by analyzing the deviation of haplotype frequencies from expected frequencies. The impact of LD on resistance to ranavirus primarily include: identifying loci associated with ranavirus resistance, precisely locating genes related to ranavirus resistance, and revealing the genetic architecture of resistance. We have included this information with a yellow background in the Materials and Methods section of the revised manuscript on page 9, “4.3. Linkage Disequilibrium and Population Structure Analysis”.

Comments 5: Please disscuss the mechanism of SNP mutation and the resistance to Ranavirus

Response 5: Thanks for your professional suggestions. In this study, we detected 2348 SNPs associated with LMBV resistance. The presence of these SNPs may affect the function of host antiviral genes or affect the host's immune response to Ranavirus. And these assumptions need to be further verification. For the detailed discussion, please refer to the yellow background in the Discussion section of the revised manuscript on page 7.

Reviewer 2 Report

Comments and Suggestions for Authors

The study reports on potential genome-wide associations regarding possible resistance to Largemouth 2 Bass Ranavirus (LMBV) in the species Micropterus salmoides.

The authors should be very clear about the gap in the literature that is covered by the findings of this study. In this respect, they must improve the Introduction by adding this information. Also, the objectives of the study must be described with clarity and conciseness.

Moreover, the authors should be their study in context: 1) current state of knowledge, 2) added value of the study, 3) who will benefit from the findings?

Methodology

5.1. Please describe clearly the control fish that were used in the study.

Results.

Visualization is excellent, but use of tables is limited, which is not supportive of the easy flow of reading of the manuscript. Please reduce text and add information in tables.

References.

The authors have missed some recent relevant references (April 2024 to today), which they can also find useful in the interpretation of their findings.

Concluding section.

1.      Is that the correct location within the manuscript?

2.      Please do not add new ideas in that section, just a sum up of the findings.

3.      Please tone down the style in the section, to bring it in line with the results.

4.      Some extrapolations can be allowed.

Overall. Improvement after extensive revision and re-evaluation.

Author Response

Comments 1: The study reports on potential genome-wide associations regarding possible resistance to Largemouth 2 Bass Ranavirus (LMBV) in the species Micropterus salmoides. The authors should be very clear about the gap in the literature that is covered by the findings of this study. In this respect, they must improve the Introduction by adding this information. Also, the objectives of the study must be described with clarity and conciseness. Moreover, the authors should be their study in context: 1) current state of knowledge, 2) added value of the study, 3) who will benefit from the findings?

Response 1: Thanks for your professional suggestions. According to the data reported by the Food and Agriculture Organization (FAO), the global production of largemouth bass in 2022 exceeded 804,000 tons. This indicates that the largemouth black bass is one of the fish species with great economic value. However, Largemouth bass ranavirus (LMBV) has caused severe economic losses in largemouth bass aquaculture. In the context of huge market demand and the impact of diseases on the development of the largemouth bass industry, breeding disease resistant strains has become the main goal of the largemouth bass industry. However, to date, molecular markers associated with disease resistance traits in largemouth bass have not been identified. Therefore, it is important to study the genetic markers associated with LMBV resistance in largemouth bass. The genetic markers obtained by GWAS in this study contribute to the molecular mechanisms of host antiviral resistance and the development of molecular marker-assisted selection in largemouth bass.

Please refer to the yellow background in the Introduction section of the revised manuscript on page 1-2.

Comments 2: Methodology

5.1. Please describe clearly the control fish that were used in the study.

Response 2: Thanks for your kindly suggestions. We provided a clearer description of the control group fish in the methodology section of our research. Please refer to the yellow background in the Materials and methods section of the revised manuscript on page 8, “4.1. Fish and LMBV Challenge”.

Comments 3: Results

Visualization is excellent, but use of tables is limited, which is not supportive of the easy flow of reading of the manuscript. Please reduce text and add information in tables.

Response 3: Thanks for your professional suggestions. Upon careful consideration, we have decided to remove the original Table 1 due to its redundancy. This decision was made to avoid overwhelming the reader with excessive information in tabular form and to maintain the focus on the most critical data. Additionally, we have created a new Table 1 that presents the essential findings in a more concise and accessible manner. This table includes the SNP genomic location, p-value, beta and PVE (proportion of variation explained). Please refer to the yellow background in the Results section of the revised manuscript on page 5, “2.4. GWAS for LMBV Resistance”.

Comments 4: References

The authors have missed some recent relevant references (April 2024 to today), which they can also find useful in the interpretation of their findings.

Response 4: Thanks for your professional suggestions. We have carefully considered the issue regarding the potential omission of recent references in our research. We acknowledge that there indeed have been some of the latest research findings not included in our analysis and discussion when writing the paper. To address this issue, we have conducted a thorough search and review of the relevant literature within the specified time period (from April 2024 to today), assessed the impact of these new publications on our research findings, and considered incorporating them into our paper. Please refer to the yellow background in the References of the revised manuscript on page 11, references 19.

Comments 5: Concluding section.

5.1 Is that the correct location within the manuscript?

Response 5.1: Thanks for your kindly suggestions. After reviewing our manuscript again, we realize that the conclusion section does need to be repositioned. We have revised the position of the conclusion section, please refer to the yellow background in the Concluding section of the revised manuscript on page 9-10.

5.2 Please do not add new ideas in that section, just a sum up of the findings.

Response 5.2: Thanks for your professional suggestions. We have re-examined and revised the conclusion section to ensure that it no longer introduces new ideas, but rather provides a concise and clear summary of the research findings. Please refer to the yellow background in the Concluding section of the revised manuscript on page 9-10.

5.3 Please tone down the style in the section, to bring it in line with the results.

Response 5.3: Thanks for your professional suggestions. We have carefully considered your suggestion and made adjustments to the style of the conclusion section to ensure consistency with the presentation of the research results. Please refer to the yellow background in the Concluding section of the revised manuscript on page 9-10.

5.4 Some extrapolations can be allowed.

Response 5.4: Thanks for your kindly suggestions. We understand that extrapolations in the conclusion section is acceptable, as it helps provide readers with a more comprehensive perspective. We have made moderate inferences in the conclusion section to demonstrate the potential impact and significance of our research findings. Please refer to the yellow background in the Concluding section of the revised manuscript on page 9-10.

Round 2

Reviewer 1 Report

Comments and Suggestions for Authors

The authors have well replied my comments, thus I recommend accept for their article.

Reviewer 2 Report

Comments and Suggestions for Authors

The authors have addressed all the issues raised and have improved the manuscript. I recommend acceptance.